# Assessing vegetation cover and valuing ecosystem services in southwestern Ethiopia: Implications for conservation

**Zenebe Ageru Yilma**[ID]*☯, **Bialfew Ashagrie Yitay**[ID]☯

Natural Resource, Mizan ATVET College, Mizan Teferi, SWERS, Ethiopia

☯ These authors contributed equally to this work.
* z.ageru@yahoo.com

**Data Availability Statement:** All relevant data are within the manuscript.

**Funding:** The author(s) received no specific funding for this work.

**Competing interests:** no conflicting interests.

## Abstract

The Bench-Sheko zone, parts of the Eastern Afromontane Biodiversity Hotspot, is characterized by its rich biodiversity. However, recent reductions in vegetation cover have drawn concern, highlighting the critical role of remote sensing in monitoring these alterations is critical. Furthermore, this study evaluates the economic value of the ecosystem services rendered by the diverse types of vegetation cover class in the area. The classification of vegetation types and measuring their ecosystem benefits are crucial for monitoring vegetation and analyzing land cover changes. estimating the value of ecosystem services is vital for environmental impact assessments, cost-benefit analyses, and creating payment schemes for these natural services. For the vegetation cover map, it uses Sentinel-2 satellite data and a Random Forest classifier using Google Earth Engine. Based on a properly chosen reference, ecosystem service assessment approaches include benefit transfer, direct market value, and the social cost of carbon. The results highlight the vegetation classes' enormous value and the services they offer. The largest value for Supporting Services (2829.3 USD ha$^{-1}$yr$^{-1}$) is found in the Remnant Forest, which makes up 30.98% of the total area. With the highest value for both cultural services (2847.7 USD ha$^{-1}$yr$^{-1}$) and regulatory services (5063.9 USD ha$^{-1}$yr$^{-1}$), the wetlands, which make up 4.35% of the total area, stand out. The total annual value of all ecosystem services provided by all vegetation classes is estimated to be 2.089 billion USD. When paired with methods for tracking and assessing changes in vegetation cover over time, high-resolution satellite images and precise classification algorithms can offer insightful information on the condition of the environment and support informed decision-making. In order to evaluate and convey to society and policymakers the advantages of vegetation cover, the value of ecosystem services is essential.

## Introduction

Vegetation is one of the main components of the terrestrial ecosystems and plays a key role in energy exchanges and in water and biogeochemical cycles of Earth [1]. Land cover is an important variable for many studies involving the Earth surface, such as climate, food security,

hydrology, soil erosion, atmospheric quality, biodiversity, and ecosystem services [2]. vegetation cover figures a land cover classification that reveals the dominant botanical composition within a specific geographical area [3].

Many landcover products are available online. Of these, the GLC200, MODIS, FAO, Copernicus Global Land Cover and Dynamic World are the main providers of land cover data globally. The GLC2000 classify the world vegetation in 22 different land cover classes, in Africa 14 land cover classes [4]. MODIS land cover product utilizes the IGBP classification system [5,6]. It uses 5 different Land cover types, for example Annual International Geosphere-Biosphere Programme (IGBP) classification or Land Cover Type 1 is important classification which classify the world in 17 classes[6]. Copernicus Global Land Cover classify the world vegetation in 5 classes and 22 land cover classes [7]. Based on global land cover created by FAO, 11 dominant land cover dataset represents the extent and distribution of the major land cover types at the global level[8]. The dynamic world land cover map classifies the world in 9 classes[9]. The Copernicus global land cover and dynamic world classify forest type as general tree cover, whereas FAO, GLC2000 and MODIS uses forest in more than one classes. A study by [10] emphasize a global trend characterized by the contraction of tree cover and the increase of agricultural and settlement areas. Between 2000 and 2020, there was a loss of 5.6% in global forest cover, while in Africa, the loss was slightly higher at 6.3%. Africa had the second highest rate of 2000–2020 net forest loss at 4.6% [10].

In sub-Saharan Africa as elsewhere, documenting and classifying vegetation has become a vital task to enable the proper assessment of endangered ecosystems [11]. Ethiopia is rich in biodiversity, topographical complexity and climate variability which jointly results in different vegetation types [12]. The Southwest part of Ethiopia is a region in the country that contains one of the counties remaining moist Afromontane Forest [13]. However, large areas of these unique forests have been converted to other land-uses [14–16]. Remote sensing combined with ground measurements plays a key role in determining loss of forest cover. Sheka, Kafa, Bench-Sheko and West Omo Zones are known for their natural forests with 60, 20 and 15% of forest cover, respectively [17]. However, the forest covers in the region at the current situation are declining both in quality and quantity at a faster rate in this decade than ever before [16].

In this region, increasing human population and the growing need for expansion of agricultural land have led to deforestation[18]. Several specific studies have been conducted which provide some indication on the rate of deforestation in Southwest Ethiopia [14,18–20]. For example, [20], estimated that the closed high forest of southwestern Ethiopia dropped from a 38% cover by 1975 to only 18% by 1997. Using aerial photographs and high- resolution satellite images, deforestation patterns between 1957 and 2007 in Southwest Ethiopia specific studies in the Jimma zone indicated a 19% decline in forest cover since 1957 [14]. According to [14], the main factors for deforestation were smallholder farmers whose mainly dependent subsistence way of lives which the author encourages shade coffee production and forest conservation could be combined. However, the study conducted in the region argued that coffee-based agroforestry practices at the cost of the natural forest is the main factor for deforestation [19]. Study in the Southwest Biosphere reserve also highlighted that the main factors for forest conversion is commercial investment and agriculture-related land expansion [21]. Ecosystem service provisioning in coffee agroforests depends strongly on the presence of adjacent forest fragments which are vital sources for propagule dispersal and ecosystem service flows which is supporting ecosystem services [22]. There have been studies utilizing Landsat satellite imagery to focus on land cover change and coffee species within biosphere reserves in the region (Kaffa, Sheka, and Yayu) [23–27]. In the recent study conducted in Shay Bench and Semen Bench Districts of Bech-Sheko Zone, the study identified seven landcover class based on Landsat satellite image. This class includes Agroforestry, Crop land, Dense Forest, Shrub land,

Settlement, Open Forest, and Wetlands [28]. Another study conducted in certain areas of Bench-Sheko has estimated the extent of vegetation cover and the ecosystem services related to carbon sequestration [29,30]. Previous studies are vital for identifying important vegetation cover and baseline information; however, they use medium resolution satellite images and miss some important vegetation classes such like savanna grassland and wetlands. This study aims to map vegetation cover in the Benchi-Sheko region, providing crucial data for sustainable ecosystem management and revealing potential ecosystem services. The monetary valuation of these services can highlight their societal and economic relevance, informing policy development, cost-benefit analyses, and ecosystem service payments.

## Methodology

**Study site.** The Bench-Sheko zone, parts of the Eastern Afromontane Biodiversity Hotspot [31], which is located between 34˚52′37″E and 35˚51′23″E longitude, and between 6˚29′51″N and 7˚14′3″N latitude (Fig 1). The study site serves as a corridor connecting the Kaffa Biosphere Reserve [32], the Sheka Forest Biosphere Reserve[33] and the Majang Forest Biosphere Reserves [34]. This area represents one of the country's remaining forests and moist Afromontane Forest [35]. The elevation was extracted from Shuttle Radar Topography

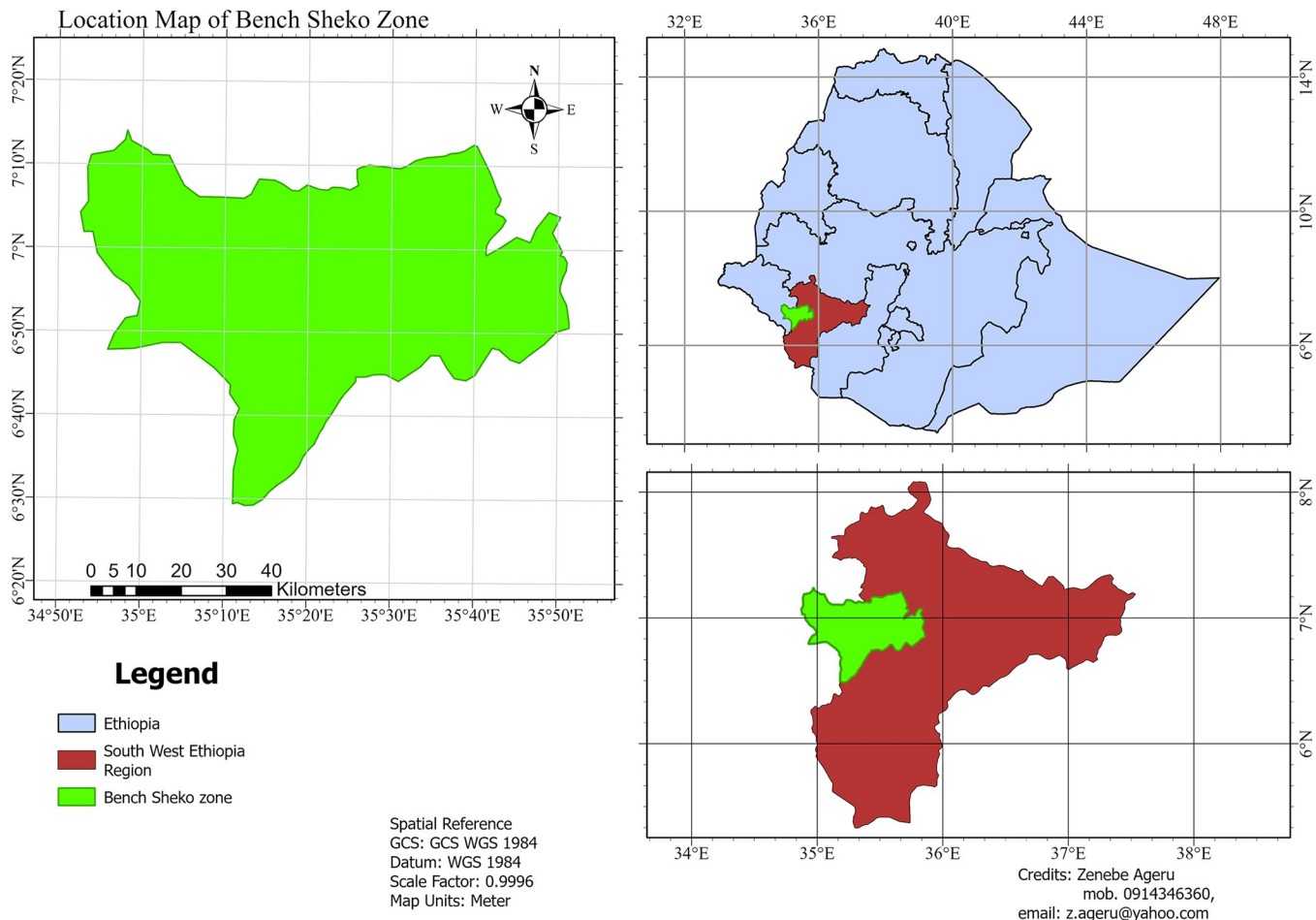

**Fig 1. Location map of the study area.** Ethiopia administrative boundaries was downloaded from GADM website (https://geodata.ucdavis.edu/gadm/gadm4.1/shp/gadm41_ETH_shp.zip) then the layout was prepared using ArcGIS pro-3.2.

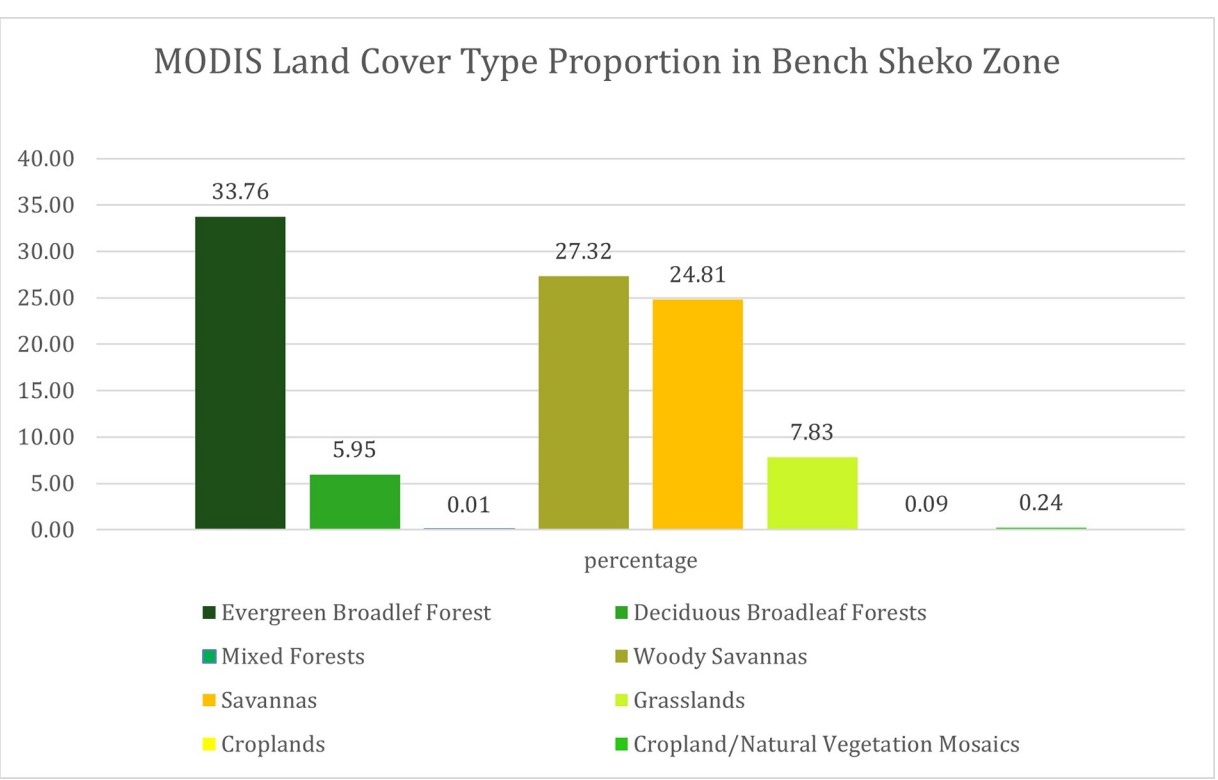

**Fig 2. MODIS land cover type proportion in bench sheko zone.**

Mission (SRTM) global digital elevation data [36] which ranges from 822 to 2371 m above sea level. Temperature and rainfall data from WorldClim 2 climate data [37] was extracted using R software [38]. Monthly mean minimum temperature and mean maximum temperature for the year 2021 were 14.49˚C and 28˚C, respectively. The average monthly rainfall for the year 2021 was 167.3 mm, resulting in an average annual rainfall of 2073.4 mm. the average annual rainfall was 1405.4 mm, with an annual average minimum rainfall of 1132 mm and an annual maximum average rainfall of 1807 mm. The average minimum temperature was 17.8˚C and the average maximum temperature was 24.1˚C.

**Field reference.** a reference point (training points) for Different vegetation class was used for Vegetation Classification inputs. The vegetation classes were determined based on different global land cover data and during reference data collection. As Shown Fig 2, MODIS Land Cover Type Yearly Global 500m [39], the study site Vegetation cover falls into 8 categories, namely Evergreen Broadleaf Forests, Deciduous Broadleaf Forests, Mixed Forests, Woody Savannas, savannas, Grasslands, crop land and Cropland/Natural Vegetation Mosaics. The identified vegetation classes were Remnant Forest, Disturbed Forest, Savannas, Grasslands, Wetlands, Coffee Plantation, Farmland, and Built-up (Table 1). These classes are representative of the different types of vegetation that can be found in the area under consideration. The Remnant Forest class refers to the remaining areas of forest that have not been relatively cleared or disturbed by human activities. The Disturbed Forest class refers to areas of forest that have been affected by human activities or plantation forest. The Savannas class refers to areas of land characterized by savanna grasses and scattered trees. The Grasslands class refers to areas of land dominated by grasses, with few or no trees. The Wetlands class refers to areas of land that are saturated with water and lush grasses. The Coffee Plantation class refers to

**Table 1. Vegetation class.**

| MODIS Land Cover Type | New vegetation classes |
|---|---|
| Deciduous Broadleaf Forests | Remnant Forest |
| Mixed Forests | Disturbed Forest |
| Woody Savannas | Savannas |
| Savannas | Savannas |
| Grasslands | Grassland and wetlands |
| crop land | farmland and Settlement |
| Cropland/Natural Vegetation Mosaics | Disturbed Forest and Coffee plantation |

areas of land that have been used for coffee plantation and shade coffee. The Farmland class refers to areas of land that are used for crop production. The Built-up class refers to areas of land that have been developed for settlement, city, and different infrastructures. The forest class in southwest Ethiopia categorized into 3 classes, namely closed high forests, slightly disturbed high forests, and heavily disturbed high forests [20]. The study identifies two classes of forests, namely Remnant Forest and Disturbed Forest.

The reference points were collected in each vegetation category. Areas representing the main vegetation types, considered to be relatively homogeneous, was collected using Global Positioning Systems (GPS). The study area covers 4721.18 square kilometer which is Larger than 1 million acres in size and have more complex maps should receive a minimum of 75 to 100 accuracy assessment sites per class [40]. In this study a minimum of 100 training points were used for both classification and validation.

**Data source.** Sentinel-2 satellite images were obtained by directly downloading the open-source data (European Space Agency [41] via Google earth engine API. Sentinel-2 is ESA's medium spatial resolution (10–60 m) super-spectral instrument aimed at ensuring data continuity for global land surface monitoring of Landsat and SPOT. Several simulation studies have been conducted that show the potential of Sentinel-2 for estimating biophysical and biochemical parameters such as leaf area index, chlorophyll and nitrogen, and spectral products such as the red edge position and NDVI time series, providing data continuity for several other operational sensors [42]. Sentinel-2 images have high potential for landscapes and forest type classification for conservation and management purposes in tropical lowland forests [43].

**Data analysis.** *Vegetation class analysis.* The study area, it worked with three distinct granule satellite images. These individual images were combined (mosaicked) into a single composite satellite image. To ensure accurate and meaningful data, several preprocessing tasks were conducted using google earth engine (Fig 3). These tasks included radiometric corrections [44–46] and the conversion of digital numbers (DN) to radiance, followed by the transformation to reflectance. Additionally, cloud and cloud shadow masking were applied to Sentinel imagery, which includes a thermal band for atmospheric correction [47]. NDVI values were calculated to evaluate Vegetation cover class.

$$\text{NDVI} = \frac{\text{Band } 8 - \text{Band } 4}{\text{Band } 8 + \text{Band } 4}$$

Where Band 8 is Near Infrared and Band 4 is red band. the value ranges from -1 to 1. The greater NDVI mean value, the healthier vegetation cover.

S2cloudless algorism was used for cloud masking and shadow detection in google earth engine [48]. Training data were generated by overlaying strategically placed points on the classified image to represent various vegetation cover classes. During data analyzing, different Supervised Classification classifiers include CART, RandomForest, NaiveBayes and SVM was

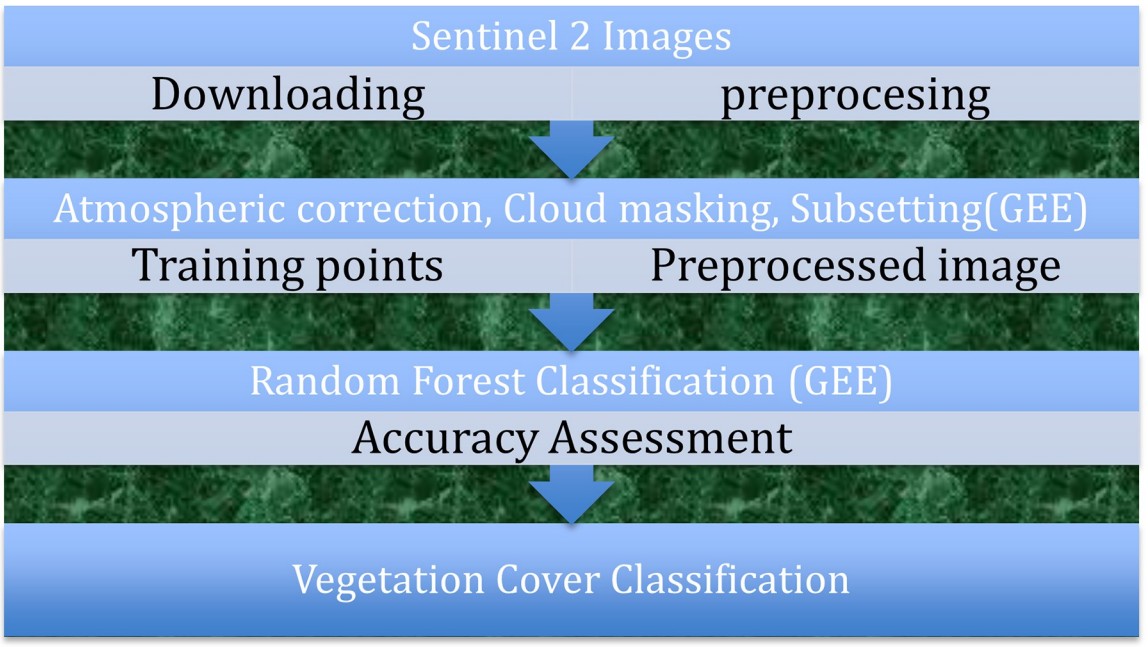

**Fig 3. Data analysis procedures.**

used, however, A random forest classifier trained on this dataset yielded best accuracy results for different vegetation types.

A random forest is a meta estimator that fits several decision tree classifiers on various sub-samples of the dataset and uses averaging to improve the predictive accuracy and control over-fitting. Trees in the forest use the best split strategy.

$$Random\ forest\ Classifier = \frac{1}{n}\sum_{i=1}^{n} T_i(x)$$

where $T_i(x)$ is the prediction of the i [th] decision tree and n is the total number of decision trees in the forest [49]. the model was trained on 80% of the data and evaluated on the remaining 20%. Final Classified vegetation cover map layout was computed using ArcGIS pro software.

### Ecosystem valuation analysis

GIS Based Benefit Transfer Method (BTM) was used for Ecosystem valuation. There are errors inherent in any function transfer. Typically, these errors are categorized as one of two types: measurement errors and transfer errors (AKA generalization error). Measurement errors are those errors that are manifest in the research related to the study site from which value is to be transferred[50]. At the first stage, the economic value for each ES service class type was calculated, using the formula [51,52].

$$ESV_k = \sum_f A_k \times VC_{kf} \tag{1}$$

$$ESV_f = \sum_k A_k \times VC_{kf} \tag{2}$$

$$ESV = \sum_f \sum_k A_k \times VC_{kf} \tag{3}$$

where $ESV_k$, $ESV_f$ and $ESV$ denote ecosystem service value of land use type k, ecosystem service value of function type f, and total ecosystem service value, respectively; $A_k$ represents the area (ha) for land use type (k); $VC_{kf}$ is the ecosystem service function value index for land use type (k) and ecosystem service function (f).

To reflect the dependence of ecosystem service value on the ecosystem service value index over time, the elasticity coefficient of economics is selected to calculate the sensitivity index [51–54].

$$CS = \left| \frac{\frac{(ESV_j) - ESV_i}{ESV_i}}{\frac{(VC_{jk} - VC_{ik})}{VC_{ik}}} \right| \tag{4}$$

initial and adjusted values, respectively; k is the land use type; and CS is coefficient of sensitivity. Adjust the value coefficient of each land use type by 50%, respectively, and then measure the change of ecosystem service value. If CS > 1, the ESV for VC is flexible; if CS < 1, the ESV for VC is lack of elasticity; CS = 1 means a complete elasticity; CS = 0 indicates a complete inelasticity. The greater the ratio is, the more important it is to the accuracy of the ecosystem service function value index.

Table 2 presents two ecosystem services, namely soil erosion control and sediment retention. These services were studied by [29]. The research was conducted in specific regions of the study area.In the study conducted by [55], the total provisioning service value of forest land, which includes honey, timber, charcoal, fuelwood, and spice, was estimated to be 196 USD ha$^{-1}$ yr$^{-1}$. The value for farmland was estimated to be higher at 638 USD ha$^{-1}$ yr$^{-1}$. These values provide a useful benchmark for estimating the value of similar ecosystems in this study (Table 4). For the purposes of this study, it is assumed that only honey production is permitted in remnant forest, while the other services are attributed to disturbed forest. As such, the value of disturbed forest is taken as 196 USD ha$^{-1}$ yr$^{-1}$, reflecting the total provisioning service value estimated by [55]. The value of remnant forest, which is primarily based on honey production, is taken as 76 USD ha$^{-1}$ yr$^{-1}$. Meanwhile, the value of farmland remains consistent with the estimate provided by [55], at 638 USD ha$^{-1}$ yr$^{-1}$. This approach allows for a nuanced understanding of the provisioning value of each land cover type in this study area, based on established research and specific assumptions about land use (Table 4).

The calculation of the monetary value of coffee production per hectare per year in USD is based on a specific formula. This formula considers both the rate of coffee production and the

**Table 2. Soil erosion control and sediment retention ecosystem service in the study area.**

| class | production cost usd/ha/yr | sediment cost usd/ha/yr | total 2018 usd/ha/yr | adjusted usd/ha/yr in 2023 |
|---|---|---|---|---|
| Remnant Forest | 4 | 64.5 | 68.5 | 84.31 |
| Disturbed Forest | 3.87 | 62.3 | 66.17 | 81.39 |
| coffee plantation | 1.08 | 17.44 | 18.52 | 22.78 |
| Savanna Grassland | 3.94 | 63.52 | 67.46 | 82.98 |

Source: [29], Estimated based on the Avoided Cost Method.

current market price of coffee.

$$GR = Rate\ of\ coffee\ production \times Present\ market\ price\ of\ coffee$$

In this case, the rate of coffee production is 0.84 ton/ha as stated by [56], and the present market price of coffee is 4.685 USD/kg.

$$NPV = GR - (GR \times 0.13)$$

The 0.13 represents the 13% production cost as a fraction of the total gross revenue, as found by [57]. Using this methodology, the value of Coffee plantation is taken as 3423.57 USD ha$^{-1}$ yr$^{-1}$ after total costs are deducted.

According to a study by [58], the provisioning ecosystem service of wetland was identified as Livestock grazing/pastures, Livestock watering, Value added through milk production, and Wetland grass for mulching. The total farmland wetland selected provisioning service a unit value of 839.98 USD ha$^{-1}$ yr$^{-1}$. Using the function transfer method and adjusting for inflation, the value of 839.98 USD ha$^{-1}$ yr$^{-1}$. in 2013 would be equivalent to 1101.62 USD ha$^{-1}$ yr$^{-1}$. in 2023. According to the standards of ecosystem service valuation implemented by the Ecosystem Services Valuation Database (ESVD), based on the study by [59] have estimated the value of provisioning services from grasslands for livestock feed in an ecosystem similar the Bale Region, Ethiopia the estimated value stands at $12.16 per hectare per year for grassland livestock provisioning services. Furthermore, the production of forage was estimated to be 0.204 [60]. a study conducted by [61], the value of provisioning services from savanna grasslands was estimated to be $41.73 per hectare per year.

Table 3 describes the coefficient of benefit transfer original valuation and adjusted coefficient. nutrient cycling, soil formation, flood regulation, water treatment, recreation, habitat service, genetic resource, and biological pest control in each vegetation class except coffee plantation was based on [62], pollination service was adopted from [63] and biological pest control for coffee plantation was adopted from [64].

In the study conducted by [30]), a comprehensive estimation of the total carbon stock was made for certain sections of a study area, specifically excluding the wetland vegetation cover. This estimation, quantified in tons per hectare, was particularly for the year 2018. For the current study, which is focused on calculating the monetary value of carbon sequestration, a more specialized approach has been adopted. This approach ensures that the calculations are based on data that is most relevant to the specific research context of this study. As a result, the methodology selected for calculating the monetary value of carbon sequestration is informed by and aligned with the findings of [30]. This decision was made considering that their research covers relevant sections of the study area, providing a robust scientific context for our calculations.

In the review of the literature, various studies have presented different estimates for the rate of total carbon stock in wetlands. For instance, [66] estimated it as 109 t C per hectare, while [67] reported a higher average of 234 t C per hectare. [68] provided an estimate for freshwater wetlands specifically, with an average organic carbon stock of 150 t C per hectare. However, for the purpose of this study, which focuses on herbaceous wetlands, the most relevant reference is [69]. They estimated the average carbon stock in herbaceous wetlands as 279.5 t C per hectare. Given that this estimate aligns most closely with the specific type of wetland under investigation in this study, it was deemed the most appropriate to use as a benchmark.

Annualized Social Cost of Carbon in 2021 ($@2015/ton) was estimated 1.47 USD/ton $CO_2$ [70]. In this study, the value of the Social Cost of Carbon (SCC) was determined to be 1.47 USD per metric ton of CO2 for the year 2021/22. The SCC represents the economic cost of the

**Table 3. Benefit transfer ecosystem service coefficients.**

| Ecosystem Service Type | Natural Forest | | Disturbed Forest | | Coffee Plantation | |
|---|---|---|---|---|---|---|
| | original value | adjusted value | original value | adjusted value | original value | adjusted value |
| Nutrient Cycling | 922.00 | 1913.70 | 361.00 | 749.28 | | |
| soil formation | 10.00 | 20.76 | 10.00 | 20.76 | | |
| Pollination Services | 623.07 | 890.32 | 668.80 | 955.67 | 939.50 | 1342.90 |
| Flood Regulation | 6.00 | 12.45 | | 4.15 | | |
| Water Treatment | 87.00 | 180.57 | 87.00 | 180.57 | | |
| Biological Pest Control | 2.00 | 4.15 | 2.00 | 4.15 | 181.25 | 248.08 |
| Medicinal Plants | 3.00 | 4.71 | 3.00 | 4.71 | | |
| Recreation | 112.00 | 232.46 | 66.00 | 136.99 | | |
| Habitat Service | 2.00 | 4.15 | 2.00 | 4.15 | | |
| Genetic Resource | 41.00 | 85.10 | 16.00 | 33.21 | | |
| Ecosystem Service Type | Savanna Grassland | | Grassland | | Farmland | | wetland | |
| | original value | adjusted value | original value | adjusted value | original value | adjusted value | original value | adjusted value |
| Nutrient Cycling | | | | | | | | |
| soil formation | 1.00 | 2.08 | 1.00 | 2.08 | | | | |
| Pollination Services | 730.00 | 1043.62 | 730.00 | 1043.62 | 192.84 | 275.68 | 196.20 | 280.47 |
| Flood Regulation | 3.00 | 6.23 | 3.00 | 6.23 | | | 15.00 | 31.13 |
| Water Treatment | 87.00 | 180.57 | 87.00 | 180.57 | | | 1659.00 | 3443.30 |
| Biological Pest Control | 23.00 | 47.74 | 23.00 | 47.74 | 24.00 | 49.81 | | |
| Medicinal Plants | | | | | | | | |
| Recreation | 2.00 | 4.15 | 2.00 | 4.15 | | | 491.00 | 1019.10 |
| Habitat Service | | | | | | | 439.00 | 911.17 |
| Genetic Resource | | | | | | | | |

Note: The provided values represent the estimated worth of various ecosystem services within different habitats, expressed in US dollars per hectare per year (USD/Ha/Yr).

damages caused by emitting one additional ton of CO2 into the atmosphere. The carbon stock in each land cover class was calculated in tons per hectare and converted to its equivalent in tons of CO2. The total carbon sequestration was then estimated for each land cover class (Table 5).

All ecosystem services presented in Tables 2–5 were summarized to Table 6 for total ecosystem service value estimation in each vegetation class and ecosystem types.

## Result

### Vegetation cover map

The land cover classification using the Google Earth Engine platform and Sentinel-2 satellite imagery successfully visualized diverse vegetation cover classes in the Bench-Sheko region (Fig 4). A cloud mask was applied to include only cloud-free pixels, and the dataset comprised Sentinel-2 imagery with 10 and 20-meter spatial resolutions, providing detailed landscape information.

Map credit: Contains modified Copernicus Sentinel satellite data, processed by google earth engine (GEE) API licensed under CC BY. GEE code to analysis the classification map can found here (https://code.earthengine.google.com/9acae293c586b66d1050dacb1c86e5b0) Training data were generated by overlaying strategically placed points on the classified image to represent various vegetation cover classes. A random forest classifier trained on this dataset yielded precise results for different vegetation types. The resulting land cover map depicted the

**Table 4. Net present value (USD ha$^{-1}$ yr$^{-1}$) of ecosystem of provisioning service in land cover class.**

| Ecosystem Type | Identified Provisioning Services | Source | Original Value (USD ha$^{-1}$ yr$^{-1}$) | Value transferred (USD ha$^{-1}$ yr$^{-1}$) |
|---|---|---|---|---|
| Remnant Forest | Honey | [65] | 76 | 85.74 |
| Disturbed Forest | Honey, Timber, Charcoal, Fuelwood, Spice | [65] | 196 | 221.11 |
| Coffee Plantation | Coffee Production | Market price and average production of coffee study site level | 3423.57 | 3423.57 |
| Wetlands | Food and fodder for livestock Livestock Watering, Milk Production, Wetland Grass for Mulching | [58] | 839.98 | 1,102.24 |
| Farmlands | Crop Production | [65] | 638 | 719.75 |
| Savanna | Livestock Feed, Forage Production | [61] | 41.73 | 41.73 |
| Grassland | livestock feed | [59] | 11.62 | 12.16 |
| | Forage | | 0.204 | 0.23 |

spatial distribution of forests, grasslands, agricultural areas, wetlands, and other vegetation classes in Bench-Sheko.

The results of this study revealed that Bench-Sheko had a rich and diverse vegetation cover, which provided various ecosystem services with significant economic value. The results also showed that the remnant natural forests were the most valuable and vulnerable areas in terms of ecosystem service provision and conservation. classification had the best overall accuracy of 82%, which means that the classified remote sensing data agreed substantially with the ground truth data. The kappa coefficient of 0.77 indicates a high level of agreement between the map and the reference data. These values are comparable to or higher than those reported in other studies that used similar methods in larger study sites. plantation coffee and disturbed forest show lower producer accuracy (0.41, 0.42) and user accuracy (0.63, 0.46) respectively (Table 7). The complexity of land cover types can vary significantly between different vegetation cover classes.

Fig 4 shows the land cover map of Bench-Sheko, which was produced using sentinel 2 imagery and supervised classification method. The map depicts the spatial distribution and extent of the 8 land cover types in the study area. The map also shows the location of the remnant natural forests, which were identified as the areas with remnant forest that were not affected by human activities such as logging, agriculture, or settlement.

These results provide valuable insights into the state of vegetation in different land cover classes in the region. The high NDVI values observed in the coffee plantation, remnant forest, and disturbed forest suggest that these areas have a high density and greenness of vegetation. In contrast, the low NDVI value observed in the savanna indicates that this area has a lower

**Table 5. Carbon stock and equivalent CO2 (ton/ha/yr).**

| Vegetation Class | Study | Carbon Stock (t C ha$^{-1}$yr$^{-1}$) | Equivalent CO2 (t Co2 ha$^{-1}$ yr-1) |
|---|---|---|---|
| Wetlands | [69] | 279.5 | 1024.83 |
| Remnant Forest | [30] | 461 | 1690.33 |
| Disturbed Forest | [30] | 396 | 1452 |
| Coffee plantation | [30] | 259.4 | 951.13 |
| Savana Grassland | [30] | 286.8 | 1051.6 |
| Grassland | [30] | 286.8 | 1051.6 |
| Farmlands | [30] | 246.5 | 903.83 |

**Table 6. Summary of adjusted ecosystem service coefficient (USD ha$^{-1}$ yr$^{-1}$).**

| Ecosystem services | Sub-ecosystem service | Natural forest | Disturbed Forest | Savanna | Grassland | Coffee plantation | wetland | farmland |
|---|---|---|---|---|---|---|---|---|
| Supporting Service | Nutrient Cycling | 1913.70 | 749.30 | | | | | |
| | Soil formation | 20.80 | 20.80 | 2.10 | 2.10 | | | |
| | pollination services | 890.60 | 955.90 | 1043.40 | 1043.40 | 1342.90 | 280.50 | 275.70 |
| | Habitat service | 4.20 | 4.20 | | | | | |
| | **sub total** | **2829.30** | **1730.20** | **1045.50** | **1045.50** | **1342.90** | **280.50** | **275.70** |
| Regulating service | Carbon Sequestration /climate Regulation | 2484.80 | 2134.40 | 1545.90 | 1545.90 | 1398.20 | 1506.50 | 1328.60 |
| | Soil Erosion Control and sediment retention | 84.30 | 81.40 | 83.00 | 83.00 | 22.80 | 83.00 | |
| | Flood regulation | 12.50 | 4.20 | 6.20 | 6.20 | | 31.10 | |
| | water treatment | 180.60 | 180.60 | 180.60 | 180.60 | | 3443.30 | |
| | biological pest control | 4.20 | 4.20 | 47.70 | 47.70 | 248.10 | | 49.80 |
| | **sub total** | **2766.40** | **2404.80** | **1863.40** | **1863.40** | **1669.10** | **5063.90** | **1378.40** |
| Provisioning service | Honey | 85.70 | | | | | | |
| | Honey, Timber, Charcoal, Fuelwood, Spice | | 221.10 | | | | | |
| | Coffee Production | | | | | 3423.60 | | |
| | Food and water for livestock and Mulching | | | | | | 1102.20 | |
| | Crop Production | | | | | | | 719.80 |
| | Livestock Feed, Forage Production | | | 41.70 | | | | |
| | livestock feed | | | | 12.20 | | | |
| | Forage | | | | 0.20 | | | |
| | Genetic Resource | 85.10 | 33.20 | | | | | |
| | **sub total** | **170.80** | **254.30** | **41.70** | **12.40** | **3423.60** | **1102.20** | **719.80** |
| Cultural service | Medicinal Plants | 4.70 | 4.70 | | | | | |
| | Recreation | 232.50 | 137.00 | 4.20 | 4.20 | | 1019.10 | |
| | Cultural | 4.20 | 4.20 | | | 4.20 | 1828.60 | |
| | **sub total** | **241.40** | **145.90** | **4.20** | **4.20** | **4.20** | **2847.70** | **0.00** |

density and greenness of vegetation. Further research and analysis may be needed to understand the factors contributing to these differences in vegetation density and greenness among different land cover classes.

Fig 5 shows the summary statistics of percentage the vegetation types in Bench-Sheko. it reveals that the dominant vegetation type was remnant forest, which covered 30.98% of the total area, followed by savanna grassland (24.47%), farmland (19.23%), coffee plantation (8%), Disturbed Forest (6.53%) grazing land (6.08%), wet land (4.35%) and settlement (0.37%). the coffee plantation in the study area had the highest mean NDVI value of 0.83, indicating a high density and greenness of vegetation. This was followed closely by the remnant forest, which had a mean NDVI value of 0.79, and the disturbed forest, with a value of 0.75. On the other hand, the savanna had the lowest mean NDVI value of 0.53, indicating a low density and greenness of vegetation. Other land cover classes in the region also showed varying degrees of vegetation density and greenness, with the grassland having a mean NDVI value of 0.71, the farmland having a value of 0.584, and the wetland having a value of 0.74.

## Ecosystem service valuation

Table 8 shows the ecosystem service monetary value of the vegetation types in Bench-Sheko, which was estimated using total carbon stock and the social cost of carbon of a study region. The table shows that the total Carbon sequestration ecosystem service value of Bench-Sheko

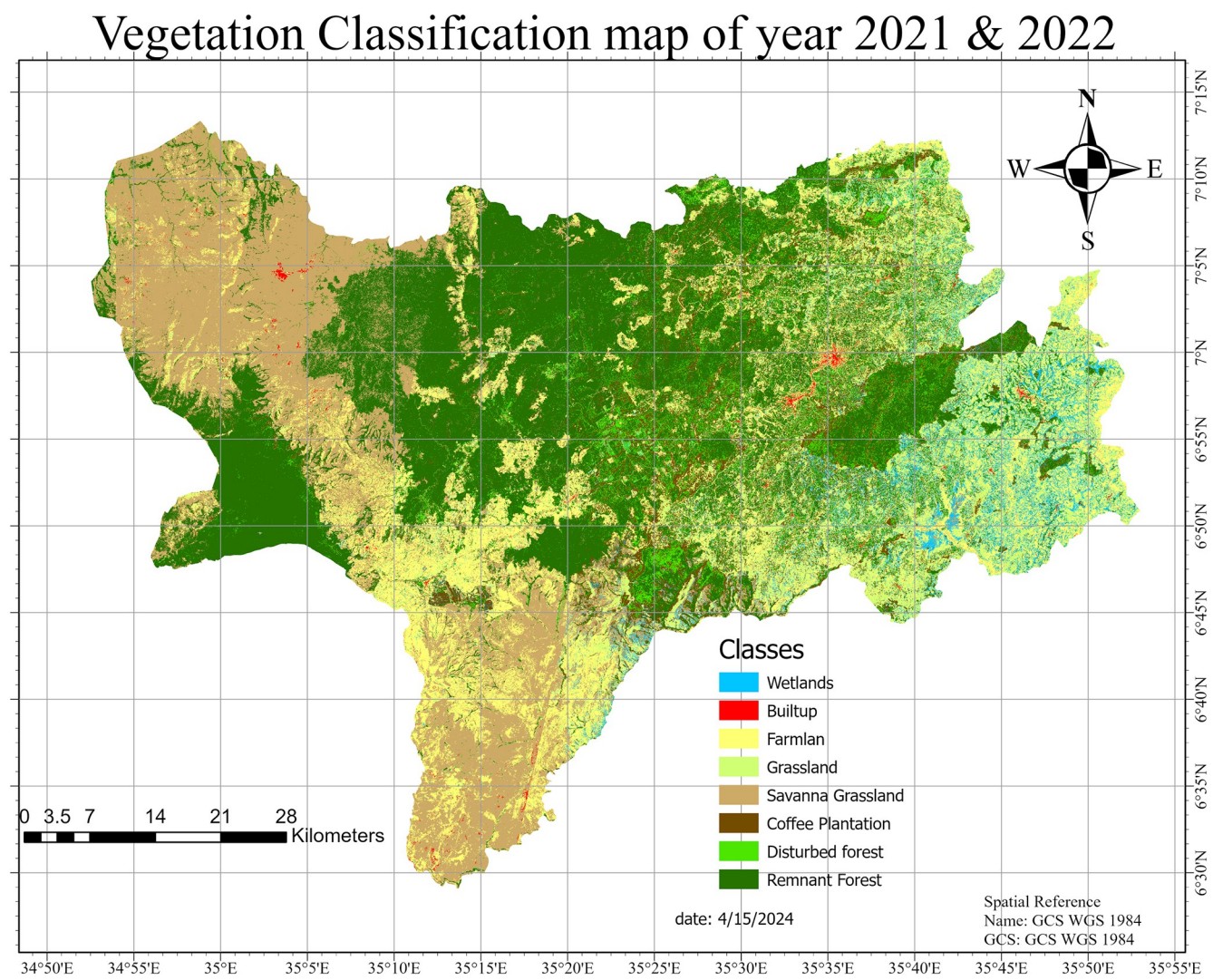

**Fig 4. Vegetation cover classification map.**

was 722.33 million USD Dollar. The table also shows that the remnant forest had the highest carbon sequestration value per hectare (2484.79 USD ha$^{-1}$ yr$^{-1}$). while the farmland had the lowest carbon sequestration value (1328.64 USD ha$^{-1}$ yr$^{-1}$).

**Table 7. Accuracy assessment for vegetation classification.**

| Kappa coefficient | Overall Accuracy | User accuracy | Vegetation Class | Producer Accuracy |
|---|---|---|---|---|
| 0.77 | 0.82 | 1.000 | Wetlands | 0.767 |
| | | 0.929 | Built-up | 0.929 |
| | | 0.720 | Farmland | 0.870 |
| | | 0.846 | Grassland | 0.943 |
| | | 0.930 | Savanna | 0.825 |
| | | 0.636 | Coffee plantation | 0.412 |
| | | 0.462 | Disturbed Forest | 0.429 |
| | | 0.792 | Remnant Forest | 0.844 |

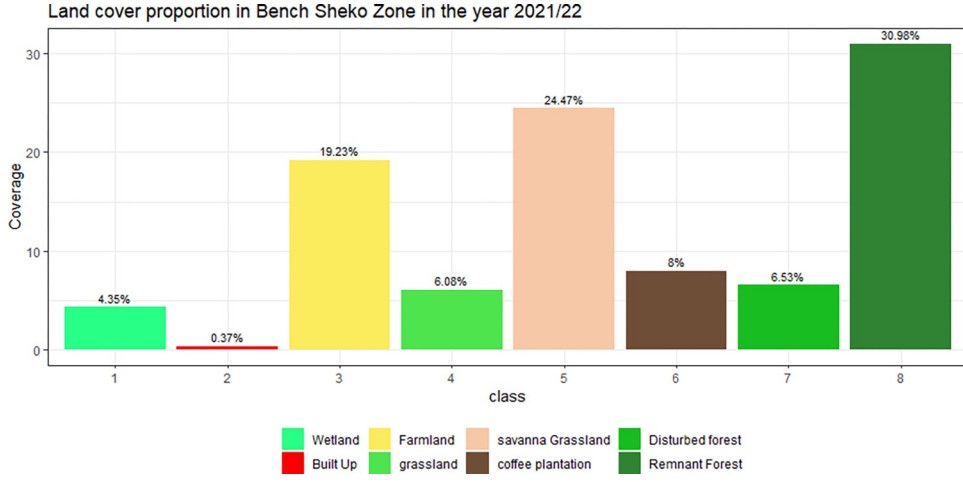

**Fig 5. Land cover proportion in bench sheko zone.**

Table 9 presents the ecosystem service value in each vegetation class. For the Supporting Service (USD ha$^{-1}$yr$^{-1}$), the highest value is observed in the Remnant Forest (2829.300), followed by the Disturbed Forest at 1730.200. The lowest value is seen in the Farmlands (275.700). In terms of the Regulating Service (USD ha$^{-1}$yr$^{-1}$), the Wetlands have the highest value (5063.900), while the Farmlands have the lowest at 1378.400. The Provisioning Service (USD ha$^{-1}$yr$^{-1}$) is highest in the Coffee plantation (3423.600), and lowest in the Savana (12.400). Lastly, for the Cultural Service (USD ha$^{-1}$yr$^{-1}$), the Wetlands again have the highest value at 2847.700, while the Farmlands have a value of 0.000, indicating no cultural service value. This data provides a comprehensive overview of the ecosystem service value across different vegetation classes. It highlights the importance of these ecosystems and the need for their conservation and sustainable use. The total ecosystem service value for each vegetation class, calculated by multiplying the ecosystem service value (USD ha$^{-1}$yr$^{-1}$) by the total area (hectares). The total ecosystem service was estimated 2.089 billion dollars.

## Discussion

Ground-based mapping presents unique challenges that complicate the effective categorization of data, as underscored by [71]. This complexity could hint at the existence of more detailed land cover types within these vegetation cover classes, in this case, coffee plantations and disturbed forests are examples. Identifying coffee plantations through remote sensing can be problematic in areas with high canopy disturbed forests, potentially leading to an

**Table 8. Total carbon sequestration monetary value of the vegetation types in Bench-Sheko.**

| Land Cover type | area (ha) | Carbon stock (ton/ha/hr) | Equivalent CO$_2$ (ton/ha/yr.) | Carbon sequestration ($ha$^{-1}$yr$^{-1}$) | Total Carbon Sequestration (million $) |
|---|---|---|---|---|---|
| Remnant Forest | 146251.59 | 461 | 1690.33 | 2484.79 | 363.40 |
| Disturbed Forest | 30832.15 | 396 | 1452.00 | 2134.44 | 65.81 |
| Coffee plantation | 37768.27 | 259.4 | 951.13 | 1398.17 | 52.81 |
| Savana Grassland | 28720.98 | 286.8 | 1051.60 | 1545.85 | 44.40 |
| Grassland | 28720.98 | 286.8 | 1051.6 | 1545.85 | 44.40 |
| wetlands | 20513.97 | 279.5 | 1024.83 | 1506.5 | 30.9 |
| Farmlands | 90778.37 | 246.5 | 903.83 | 1328.64 | 120.61 |

**Table 9. Total ecosystem service estimation in each ecosystem category and vegetation class.**

| Land Cover type | Area (ha) | Supporting service (USD ha$^{-1}$yr$^{-1}$) | Total Supporting service (Million $) | Regulating service (USD ha$^{-1}$yr$^{-1}$) | Total Regulating service (Million $) | Provisioning service (USD ha$^{-1}$yr$^{-1}$) | Total Provisioning service (Million $) | Cultural service (USD ha$^{-1}$yr$^{-1}$) | Total Cultural service (Million $) |
|---|---|---|---|---|---|---|---|---|---|
| Remnant Forest | 146929.5 | 2829.300 | 415.708 | 2766.400 | 406.466 | 170.800 | 25.096 | 241.400 | 35.469 |
| Disturbed Forest | 28241.54 | 1730.200 | 48.864 | 2404.800 | 67.915 | 254.300 | 7.182 | 145.900 | 4.120 |
| Coffee plantation | 41893.27 | 1342.900 | 56.258 | 1669.100 | 69.924 | 3423.600 | 143.426 | 4.200 | 0.176 |
| Savana | 113093.5 | 1045.500 | 118.239 | 1863.400 | 210.738 | 41.700 | 4.716 | 4.200 | 0.475 |
| Grassland | 26716.23 | 1045.500 | 27.932 | 1863.400 | 49.783 | 12.400 | 0.331 | 4.200 | 0.112 |
| wetlands | 18396.11 | 280.500 | 5.160 | 5063.900 | 93.156 | 1102.200 | 20.276 | 2847.700 | 52.387 |
| Farmlands | 94844.59 | 275.700 | 26.149 | 1378.400 | 130.734 | 719.800 | 68.269 | 0.000 | 0.000 |

The total ecosystem service value of different land use types with respect to the coefficient of sensitivity changes of ecosystem service value is 1, which indicates that the total ecosystem service value is relatively elastic.

underestimation of coffee plantations. Despite these challenges, the classification of high-resolution satellite scenes can benefit from the integration of multiple features, as suggested by [72]. In various studies, Coffee plantations and Disturbed Forests have been categorized differently. For example, a study conducted by [28] classifies lands covered by coffee plantations, spices, woodlots, and fruit trees grown within homesteads as agroforestry within the study area. In specific regions of the study area, an additional investigation [30] classifies the same land cover types as in the study. For instance, while referring to them as 'disturbed forest' and 'coffee plantation,' the other study categorizes them as 'semi-forest' and 'coffee investment.' These classifications exhibit a notable similarity between the two land cover classes. In both studies, Dense Forest [28] and Natural Forest [30], the classification of Remnant Forest is consistent. The forest coverage in the Semen Bench and Shay Bench districts of the Bench Sheko zone was estimated to be 15.2% according to a study [28]. This estimate is nearly in agreement with the coverage observed in the specified study area, which stands at 17.6%. However, there is a discrepancy regarding the coverage of farmland and wetland. The study contends that wetlands account for 15.14% and farmland for 34.57%, whereas the reported figures are 4.25% for wetlands and 23.03% for farmland. In another study conducted in the Sheko, South Bench, and Guraferda districts [30], where most of the remnant forest cover is concentrated in Bench Sheko, the findings indicate that the remnant forest covers 55.7% of the area. Remarkably, this aligns with the results from the specified study, which reported a remnant forest coverage of 63.65% in the study. The consensus is clear: the remnant forest occupies a significant portion of the landscape, exceeding the 55% threshold. Furthermore, the study sheds light on farmland distribution. Specifically, it identifies farmland coverage at 7.9% in the specified area and 10.6% in this study, which is nearly similar.

Wetlands comprise 4.35% of the study area, specifically confined in the Shay Bench Districts of Bench Sheko Zone (Fig 5), yet their classification has been frequently overlooked in global land cover datasets covering the region. Notably, wetlands are not distinctly represented in widely used global land cover maps such as the Moderate Resolution Imaging Spectroradiometer (MODIS) [39] and the European Space Agency (ESA) world land cover maps [73] after clipping to the study area. Unlike these datasets, the Food and Agriculture Organization (FAO) classification appropriately identifies wetlands as a distinct land cover class [74]. This implies that global landcover maps remain valuable datasets to be used as a first reference and

baseline data source, it routinely shows spatial inconsistency when used in relation to one another [75]. These discrepancies arise due to their large coverage and some of the land cover categories are small-scale and highly complex [76]. For instance, distinguishing wetlands and grasslands can be challenging due to their intricate spatial arrangements.

According to a study by [77], it was discovered that wetlands can sequester carbon at rates that are 30 to 50 times higher than those of forests. Interestingly, despite this high sequestration rate, wetlands were found to have a lower carbon sequestration value in comparison to other vegetation classes, except for farmland and coffee plantations. Adding another layer of complexity, wetlands were observed to provide the highest regulatory service when compared to all other vegetation classes.

As indicated in Table 7, the Remnant Forest exhibited a lower provisioning service than coffee plantation and disturbed forest and the highest climate regulation service. This finding suggests that the economic valuation of forest provisioning ecosystem services and climate regulation can play a crucial role in elucidating trade-offs among conflicting environmental, social, and economic objectives during the development and implementation of policies, as well as in enhancing ecosystem management to preserve biodiversity [78]. For instance, permitting certain provisioning services such as charcoal, timber, and raw materials or converting remnant forest to coffee plantation may result in a decrease in climate regulation ecosystem services or an increase in the value of provisioning service at the expense of regulatory and supporting ecosystem services [30].

In the context of this research, the cultural ecosystem services (CES) value for farmlands was initially found to be zero using the benefit transfer method. This finding presents a contradiction when compared with empirical observations and the inherent understanding of the role of farmlands [79]. A study found that agricultural landscapes generally have a stronger capacity to provide supply services, but a weaker capacity for regulation services, cultural services, and support services [80]. This suggests that the emphasis on the cultural services of farmlands could potentially compromise their ability to provide other important services, such as regulation and support services. Moreover, recreational activities, which are a form of cultural service, can also create negative feedback, affecting the ecosystem, biodiversity, and other CES [81]. For example, tourism activities in farmlands can lead to degradation of the land and disturbance of wildlife, thereby affecting the ecosystem's ability to provide other services. Therefore, while it's important to recognize and value the cultural services provided by farmlands, it's equally crucial to consider the potential trade-offs and strive for a balance that ensures the sustainability of all ecosystem services. This calls for integrated land management strategies that consider all ecosystem services and aim for their optimal provision.

In the Sheka Biosphere Reserve, Southwest Ethiopia, [82] estimated the total ecosystem services value coefficients in USD ha$^{-1}$yr$^{-1}$ for wetland (8103.5. In the present study, a comparable estimate for wetland (9294.3) was obtained. However, for Remnant Forest, coffee plantation and farmland total ecosystem services value coefficients is unlikely to lower estimates by similar study, [82]. This implies that the total ecosystem services in original estimates in the past year may use to estimate the present values. In the Kaffa Biosphere Reserve, [83] estimated the total ecosystem services value coefficients in USD per hectare per year for remnant forest (5382). Our study yields a comparable estimate for remnant forest (6007.9). However, the other land cover class total ecosystem services estimated by the study significantly exceed this finding. The observed disparities in total ecosystem services across various vegetation cover classes in different studies may arise from variations in vegetation composition between study sites and policy sites [84]. To address these discrepancies, it is advisable to conduct further research using both direct and indirect valuation methods. Such investigations would help reconcile variations in ecosystem coefficient values and enhance the precision of our assessments.

## Conclusion

The power of high-resolution satellite imagery in providing a more detailed and accurate representation of vegetation classes in Bench-Sheko. The remnant forests, which have the highest coverage among all vegetation classes, are identified as crucial areas for ecosystem service value. Wetlands, despite their smaller coverage, also provide remarkable ecosystem service value. This highlights the importance of including all significant vegetation classes in land cover classifications, regardless of their size. In addition to these, coffee plantations also play a significant role. It provides the highest provisioning ecosystem service, contributing significantly to the local economy and livelihoods. However, it's important to manage these plantations sustainably to prevent any adverse impacts on the environment and other ecosystem services. The study advocates for the sustainable utilization of vegetation classes in a manner that ensures mutual benefits. Specifically, it proposes the promotion of activities such as bee production, seed collection, and the use of non-consumptive resources utilization within remnant forest classes. This approach can harmonize the trade-off between provisioning and other types of ecosystem services, thereby ensuring the sustainable use of these valuable natural resources. Overall, the study underscores the importance of using advanced tools and methods in environmental studies. It also emphasizes the need for sustainable practices that balance economic benefits with ecosystem conservation. The insights gained from this study can guide policy decisions and contribute to the sustainable management of the rich and diverse vegetation cover in Bench-Sheko.

## Author Contributions

**Conceptualization:** Zenebe Ageru Yilma, Bialfew Ashagrie Yitay.

**Data curation:** Zenebe Ageru Yilma, Bialfew Ashagrie Yitay.

**Formal analysis:** Zenebe Ageru Yilma.

**Investigation:** Zenebe Ageru Yilma.

**Methodology:** Zenebe Ageru Yilma, Bialfew Ashagrie Yitay.

**Project administration:** Zenebe Ageru Yilma.

**Resources:** Zenebe Ageru Yilma, Bialfew Ashagrie Yitay.

**Software:** Zenebe Ageru Yilma.

**Supervision:** Zenebe Ageru Yilma, Bialfew Ashagrie Yitay.

**Validation:** Zenebe Ageru Yilma.

**Visualization:** Zenebe Ageru Yilma.

**Writing – original draft:** Zenebe Ageru Yilma.

**Writing – review & editing:** Zenebe Ageru Yilma.

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
