## [Decision Letter · Decision Letter 0]

1 Apr 2024

PONE-D-24-00098Assessing Vegetation Cover and Valuing Ecosystem Services in Southwestern Ethiopia: Implications for Conservation.PLOS ONE

Dear Dr. YILMA,

Thank you for submitting your manuscript to PLOS ONE. After careful consideration, we feel that it has merit but does not fully meet PLOS ONE’s publication criteria as it currently stands. Therefore, we invite you to submit a revised version of the manuscript that addresses the points raised during the review process.

We look forward to receiving your revised manuscript.

Kind regards,

Mattias Gaglio, PhD

Academic Editor

PLOS ONE

4. In the online submission form, you indicated that [Data Access / Ethics Committee (contact via z.ageru@yahoo.com) for researchers who meet the criteria for access to confidential data.]. 

5. Please ensure that you include a title page within your main document. You should list all authors and all affiliations as per our author instructions and clearly indicate the corresponding author.

6. Please amend your list of authors on the manuscript to ensure that each author is linked to an affiliation. Authors’ affiliations should reflect the institution where the work was done (if authors moved subsequently, you can also list the new affiliation stating “current affiliation:….” as necessary).

7. We note that Figure 1 and 4 in your submission contain [map/satellite] images which may be copyrighted. All PLOS content is published under the Creative Commons Attribution License (CC BY 4.0), which means that the manuscript, images, and Supporting Information files will be freely available online, and any third party is permitted to access, download, copy, distribute, and use these materials in any way, even commercially, with proper attribution. For these reasons, we cannot publish previously copyrighted maps or satellite images created using proprietary data, such as Google software (Google Maps, Street View, and Earth). For more information, see our copyright guidelines: http://journals.plos.org/plosone/s/licenses-and-copyright.

1. You may seek permission from the original copyright holder of Figure 1 and 4 to publish the content specifically under the CC BY 4.0 license.  

Reviewers' comments:

Reviewer's Responses to Questions

**Comments to the Author**

1. Is the manuscript technically sound, and do the data support the conclusions?

Reviewer #1: Yes

Reviewer #2: Partly

2. Has the statistical analysis been performed appropriately and rigorously? 

Reviewer #1: Yes

Reviewer #2: N/A

3. Have the authors made all data underlying the findings in their manuscript fully available?

Reviewer #1: Yes

Reviewer #2: Yes

4. Is the manuscript presented in an intelligible fashion and written in standard English?

Reviewer #1: Yes

Reviewer #2: No

5. Review Comments to the Author

Reviewer #1: 1.Abstract

This part briefly introduces the whole process of the study, and clearly states the results and views of the study. Although the research is very valuable, the background and purpose of the research are not stated. Nowadays, there are many scientific problems that need to be solved in the world or in Ethiopia. However, I do not know from the abstract what kind of international environment the author conducts the research based on, what kind of problems to deal with and what kind of purpose he started it. It is recommended to start this section with a 1-2 sentence brief explanation.

It is not accurate to use the expression "biodiversity" in the first sentence of the abstract. Firstly, biodiversity includes the diversity of animals and plants, and this paper does not explore the diversity of animals. Secondly, this paper discusses the economic value of different vegetation cover and ecosystem services, but does not mention the role or relationship of plant diversity characteristics, and does not involve the calculation and analysis of plant diversity indicators of different vegetation cover types. Therefore, it is suggested that the author reconsider the expression method.

2. Introduction

This section explains the reasons for studying the vegetation cover of Ethiopia and the value of its ecosystem services. A series of anthropogenic factors, such as urban expansion and deforestation, have resulted in the degradation and loss of forests in the region. Ethiopia's rich flora and fauna have been damaged to varying extent. In this process, the monitoring system based on satellite remote sensing plays a great role, and this study is also based on this. So at present, have some scholars conducted research on this level? What are the latest research results in this field in the world? Do their results provide the basis for this study? And what does this study do on the basis of previous studies? The authors are requested to supplement the content of the review in appropriate places.

3. Methodology

In the part of the Study site, I have clearly understood the basic information of Bechi-Sheko in Ethiopia. Please briefly explain why you choose this region instead of other regions.

In the part of Data Analysis, I have been clear about the research method and data processing process, but why do we use this method in this study? What are the advantages of similar methods? Were the same types of methods compared and screened before this study began?

4. Result

This part fully introduces the research results and the process of drawing production, and provides necessary explanations and explanations for the drawing. However, the significance of the two indicators of overall accuracy and kappa coefficient is widely known. It is suggested to properly check the description of the Result part and simplify the explanation of the redundant part. Avoid over-description.

The image of Figure4.5 is not high in definition, please provide a clear enough image of the result.

5. Discussion

The first paragraph of this part should be the content of the Result part, please merge and adjust.

In the third paragraph of this section, the view that "wetland classification has been neglected" is confusing. Due to differences in techniques and classification criteria, different studies produce different classifications, which are equal. In your study, the result that wetlands occupy 4.35% of the study area only shows that the classification of wetlands can be obtained using your method and criteria, and its ecosystem service value is correspondingly very limited. Since the area occupies a small proportion and the contribution is limited, how can we prove that the humidity area is essential? In addition, MODIS and ESA cover the whole world, and the scope of this study is small, so I doubt this conclusion, please further explain.

In the fourth paragraph of this part, since the choice of evaluation method will affect the result, it is necessary to explain the reasons for the choice of method and the choice of region. Please adjust it in accordance with the issues in section "3. Methodology".

Reviewer #2: The study has great potential and is very important especially given the area where the study was carried out. Despite this, the study needs major revisions especially in terms of structure and writing. Not respecting the guidelines makes it difficult to give precise indications because the number of lines and/or the number of pages are missing, however, for example, the paragraph "Data Analysis" is difficult to read and understand with some points where it appears to be simple annotations (i.e. "Radiometric conversions and correction."). Not making all acronyms explicit makes reading difficult. In the results, the NDVI value is mentioned but not shown. The discussions also seem quite messy. I propose a discussion for each service analysed in relation to the type of land cover. In the conclusions highlight more the importance and maybe give some suggestions in terms of conservation.

The English should be revised as well as the spelling part. Sentences should always begin with a capital letter.

6. PLOS authors have the option to publish the peer review history of their article (what does this mean?). If published, this will include your full peer review and any attached files.

Reviewer #1: No

Reviewer #2: No

---

## [Author Response · Author response to Decision Letter 0]

18 Apr 2024

Respond to reviewers:

Reviewer #1: 1.Abstract

This part briefly introduces the whole process of the study, and clearly states the results and views of the study. Although the research is very valuable, the background and purpose of the research are not stated. Nowadays, there are many scientific problems that need to be solved in the world or in Ethiopia. However, I do not know from the abstract what kind of international environment the author conducts the research based on, what kind of problems to deal with and what kind of purpose he started it. It is recommended to start this section with a 1-2 sentence brief explanation.

It is not accurate to use the expression "biodiversity" in the first sentence of the abstract. Firstly, biodiversity includes the diversity of animals and plants, and this paper does not explore the diversity of animals. Secondly, this paper discusses the economic value of different vegetation cover and ecosystem services, but does not mention the role or relationship of plant diversity characteristics, and does not involve the calculation and analysis of plant diversity indicators of different vegetation cover types. Therefore, it is suggested that the author reconsider the expression method.

Based on the comments Provided, we have added the international environment “ the area is parts Eastern Afromontane Biodiversity Hotspot” and the purpose also stated, final we have changed the word biodiversity 

2. Introduction

This section explains the reasons for studying the vegetation cover of Ethiopia and the value of its ecosystem services. A series of anthropogenic factors, such as urban expansion and deforestation, have resulted in the degradation and loss of forests in the region. Ethiopia's rich flora and fauna have been damaged to varying extent. In this process, the monitoring system based on satellite remote sensing plays a great role, and this study is also based on this. So at present, have some scholars conducted research on this level? What are the latest research results in this field in the world? Do their results provide the basis for this study? And what does this study do on the basis of previous studies? The authors are requested to supplement the content of the review in appropriate places.

We have put references about land cover of the world and some recent findings of parts of the study area and similar environment in adjacent areas. 

3. Methodology

In the part of the Study site, I have clearly understood the basic information of Bechi-Sheko in Ethiopia. Please briefly explain why you choose this region instead of other regions.

In the part of Data Analysis, I have been clear about the research method and data processing process, but why do we use this method in this study? What are the advantages of similar methods? Were the same types of methods compared and screened before this study began?

We have stated why we choose the area and why the method is preference “ During data analysing, different Supervised Classification classifiers include CART, RandomForest, NaiveBayes and SVM was used, however, A random forest classifier trained on this dataset yielded best accuracy results for different vegetation types.”

4. Result

This part fully introduces the research results and the process of drawing production, and provides necessary explanations and explanations for the drawing. However, the significance of the two indicators of overall accuracy and kappa coefficient is widely known. It is suggested to properly check the description of the Result part and simplify the explanation of the redundant part. Avoid over-description.

The image of Figure4.5 is not high in definition, please provide a clear enough image of the result.

Based on the comment we have corrected the image in accordance with plos one requirement and rewrite some redundant parts

5. Discussion

The first paragraph of this part should be the content of the Result part, please merge and adjust.

Corrected 

In the third paragraph of this section, the view that "wetland classification has been neglected" is confusing. Due to differences in techniques and classification criteria, different studies produce different classifications, which are equal. In your study, the result that wetlands occupy 4.35% of the study area only shows that the classification of wetlands can be obtained using your method and criteria, and its ecosystem service value is correspondingly very limited. Since the area occupies a small proportion and the contribution is limited, how can we prove that the humidity area is essential? In addition, MODIS and ESA cover the whole world, and the scope of this study is small, so I doubt this conclusion, please further explain.

Based on the comment, we put further explanation to avoid misunderstanding and to clear the potentials of world land cover and their limitation as well. The wetland coverage in the study area is not faily distributed in all parts of the study area, most of wetland cover found in shay bench districts of Bench Sheko, therefore 4.35% is big enough in such case. The study emphases vegetative wetlands may confuse with grasslands during classification. 

In the fourth paragraph of this part, since the choice of evaluation method will affect the result, it is necessary to explain the reasons for the choice of method and the choice of region. Please adjust it in accordance with the issues in section "3. Methodology".

Corrected 

Reviewer #2: The study has great potential and is very important especially given the area where the study was carried out. Despite this, the study needs major revisions especially in terms of structure and writing. Not respecting the guidelines makes it difficult to give precise indications because the number of lines and/or the number of pages are missing, however, for example, the paragraph "Data Analysis" is difficult to read and understand with some points where it appears to be simple annotations (i.e. "Radiometric conversions and correction."). Not making all acronyms explicit makes reading difficult. In the results, the NDVI value is mentioned but not shown. 

Based on the comment, we have done many editorial problems, for example Reference style, line spaces . we rewrite the data analysis grammar problem part. 

The discussions also seem quite messy. I propose a discussion for each service analysed in relation to the type of land cover. In the conclusions highlight more the importance and maybe give some suggestions in terms of conservation.

The English should be revised as well as the spelling part. Sentences should always begin with a capital letter.

The discussion is rewritten based on the comments.

---

## [Decision Letter · Decision Letter 1]

14 May 2024

Assessing Vegetation Cover and Valuing Ecosystem Services in Southwestern Ethiopia: Implications for Conservation.

PONE-D-24-00098R1

Dear Dr. YILMA,

We’re pleased to inform you that your manuscript has been judged scientifically suitable for publication and will be formally accepted for publication once it meets all outstanding technical requirements.

Kind regards,

Mattias Gaglio, PhD

Academic Editor

PLOS ONE

Additional Editor Comments (optional):

Reviewers' comments:

Reviewer's Responses to Questions

**Comments to the Author**

1. If the authors have adequately addressed your comments raised in a previous round of review and you feel that this manuscript is now acceptable for publication, you may indicate that here to bypass the “Comments to the Author” section, enter your conflict of interest statement in the “Confidential to Editor” section, and submit your "Accept" recommendation.

Reviewer #2: All comments have been addressed

2. Is the manuscript technically sound, and do the data support the conclusions?

Reviewer #2: Yes

3. Has the statistical analysis been performed appropriately and rigorously? 

Reviewer #2: N/A

4. Have the authors made all data underlying the findings in their manuscript fully available?

Reviewer #2: Yes

5. Is the manuscript presented in an intelligible fashion and written in standard English?

Reviewer #2: Yes

6. Review Comments to the Author

Reviewer #2: (No Response)

7. PLOS authors have the option to publish the peer review history of their article (what does this mean?). If published, this will include your full peer review and any attached files.

Reviewer #2: No

---

## [Editor Report · Acceptance letter]

27 May 2024

PONE-D-24-00098R1 

PLOS ONE

Dear Dr. YILMA, 

I'm pleased to inform you that your manuscript has been deemed suitable for publication in PLOS ONE. Congratulations! Your manuscript is now being handed over to our production team.

Kind regards, 

on behalf of

Dr. Mattias Gaglio 

Academic Editor

PLOS ONE